

# Simulated evidence for ice over 1Ma in the Dome A region, East Antarctica

Dong Li[1,2], Xueyuan Tang[1,2,3,4], Ailsa Chung[5,6], Frédéric Parrenin[7], Bo Sun[1]

[1]School of Oceanography, Shanghai Jiao Tong University, Shanghai 200230, China
[2]Key Laboratory of Polar Science of Ministry of Natural Resources (MNR), Polar Research Institute of China, Shanghai 200136, China
[3]Key laboratory of Polar Atmosphere-ocean-ice System for Weather and climate, Ministry of education, department of Atmospheric and Oceanic Sciences & institute of Atmospheric Sciences, Fudan University, Shanghai 200438, China
[4]Ocean College, Zhejiang University, Zhoushan 316021, China
[5]University of Bern, Physics Institute, Climate and Environmental Physics, Bern, Switzerland
[6]Oeschger Centre for Climate Change Research, University of Bern, Bern, Switzerland
[7]Univ. Grenoble Alpes, CNRS, INRAE, IRD, Grenoble INP, IGE, 38000 Grenoble, France

Correspondence to: Xueyuan Tang (tangxueyuan0369@sjtu.edu.cn)

**Abstract.** The search for Antarctic ice-core records older than 1Ma is crucial for advancing our understanding of global climate change. However, basal melting and complex internal deformation within the Antarctic Ice Sheet make it extremely challenging to obtain continuous ice core records exceeding 1 Ma. In this study, we integrate ice core observations, radar transect constraints, and a pseudo steady state modeling framework to construct a high precision age model and evaluate the age of old ice in the Dome A region. Our simulations indicate extremely low basal melting (0.14 mm/yr) at Kunlun Station, ,

and a maximum ice age of 1737 ± 223 ka, corresponding to low age density of 8.2 kyr/m. Considering the presence of basal melting, we adopt the age at 200 m above the bedrock as the maximum ice age, which yields an age 909 ± 113 ka and age density of 2.5 kyr/m, which is far lower than the maximum age density of 20 kyr/m required for high resolution climate reconstructions. Finally, we identify three potential locations where ice older than 1Ma may be preserved, with the area north of Kunlun Station exhibiting both older ice and higher age density. This work demonstrates the generality and transferability

of the Isoinv1D model, providing important support for Chinese deep ice core drilling program at Dome A and accelerating the IPICS effort to locate and recover million-year-old ice.




## 1 Introduction


The Mid-Pleistocene Transition (MPT) refers to the Quaternary period's interglacial glacial cycle, which shifted from a 40 ka cycle in the Early Pleistocene to a 100 ka cycle in the Late Pleistocene, accompanied by an increase in the amplitude (e.g., global temperature, glacier volume, etc.) of the glacial cycle (Clark et al., 2006). Research suggests this may be related to the effects of greenhouse gases such as $CO_2$ at the time (Willeit et al., 2019). Analysis of gases extracted from Antarctic ice cores

revealed that the millennial-scale variations in atmospheric methane recorded in the EPICA Dome C (EDC) ice core may be related to global climate change (Jouzel et al., 2007). This is crucial for understanding climate change in the MPT and also for future climate simulations. Among the five priority projects of the International Partnership in Ice Core Sciences (IPICS), the primary goal is to find the oldest ice cores, which is also the greatest challenge facing the study of the MPT through ice cores. Of the existing ice cores in Antarctica, the oldest analyzed ice core sample is the EDC in East Antarctica, with a maximum

age of approximately 800,000 years (Bouchet et al., 2023). The recent Beyond EPICA project aims to drill ice cores older than 1.5 million years. Significant progress has been made, with an ice core drilled to around 2,800 meters deep at Little Dome C (LDC) in Antarctica. Preliminary estimates suggest that the preserved climate record is at least 1.2 million years old (Chung et al., 2023). Analysis of the Vostok ice core indicates that a layer of complexly deformed ice exists at the bottom 228 m of the ice core, near the bedrock, resulting in sub-meter-scale folding and mixing of the ice. This is known as the basal layer

(Souchez et al., 2002). The current major challenge with drilling deep ice cores is that basal pressure can cause deformations such as shear reversal in the bottom ice, leading to confusion in temporal information and the inability to obtain a clear paleoclimate signal (Tison et al., 2015). Furthermore, basal melting makes it difficult to preserve older ice. Therefore, finding a continuous ice core record that has not melted or has experienced low melting and is undisturbed is a major challenge.

In the previous European EPICA project, researchers (Chung et al., 2023; Parrenin et al., 2017) investigated the ice age

in the Dome C region by combining radar surveys with 1D models. Their simulations indicated the presence of stagnant ice in the LDC, with a maximum age of $1.45 \pm 0.16$ Ma. Simulated ages exceeded 2 Ma at another more promising old ice location, the NP, an area located 10-15km north of EDC. Wang et al. (2023b) used the same model combined with radar surveys to investigate the distribution of old ice in the Dome Fuji region, identifying four potential drilling locations with simulated maximum ice ages exceeding 1.5 Ma. Where basal simulations indicate melting, the simulated age at bedrock depth is taken

as the maximum ice age. In contrast, where basal simulations indicate freezing, the maximum ice age is defined as ice characterized by an age density of 20 kyr/m. They also compared the simulated basal states with those of subglacial lakes, revealing that most of these locations exhibited melting conditions. The Dome A region, located at the highest point in Antarctica, is considered to harbor relatively old ice (Frezzotti et al., 2005). Ice core drilling is currently underway at Kunlun Station. Located at $80°25'01''S, 77°06'58''E$, it has an average annual temperature of -58.5°C, an ice thickness of

approximately 3000 m, and an extremely low surface mass balance (Hou et al., 2007). China has previously drilled ice cores near Kunlun Station, but only a shallow ice core of ~800 m has been obtained (Hu et al., 2021). Van Liefferinge and Pattyn (2013) showed that the best locations to look for old ice are where the heat flux is low and the ice thickness is relatively small,





because too thick ice can lead to temperate basal conditions. The low accumulation rate at Kunlun Station results in slow ice formation, and the extremely low temperatures effectively suppress basal melting, factors that favor the preservation of old ice. Previous studies by Sun et al. (2014) using a full Stokes coupled heat equation model showed that, in the absence of basal melting, ice ages at 95% depth in the Dome A region could be constrained to less than 1.5 Ma. High-resolution ice cores dating to 600-700 ka are possible at Kunlun Station. Using the same methodology and the best available ice fabric, Zhao et al. (2018) found that within 400 m of Kunlun Station, 1 Ma ice could exist 200 m above the bedrock. Their models are constrained by surface velocity measurements and prescribed geothermal flux (GHF). However, GHF is spatially variable, and surface velocity observations are sparse. These factors will lead to greater uncertainty in age of deep ice. By contrast, our approach offers two advantages: (1) It is constrained by dated IRHs, which have a stronger constraints on age-depth structure; and (2) The model is well established and has been successfully applied in multiple regions, especially at LDC. These factors will strongly supportf the progress of the Dome A ice core project,DK-1 (Zhang et al., 2014). In this paper, we use the 1D pseudo-steady-state model of Chung et al. (2023) and the dated radar line constraints from the 21st Chinese National Antarctic Research Expedition (CHINARE 21) to simulate the age-depth relationship and basal melting or stagnation state in the Dome A region. The accuracy of the simulation results is verified by comparing them with the actual observed ages. At the same time, our simulation results are compared with the research results based on 3D models by Zhao et al. (2018) and Sun et al. (2014) to explore the most reasonable oldest ice drilling locations.

## 2 Data processing

### 2.1 CHINARE 21

The data used in this paper comes from radar echo sounding data collected by the CHINARE 21 in the Dome A region during 2004/2005 (Wang et al.,2016; Bingham et al., 2024), which we refer to as the C21 radar survey. The radar system used a dual-frequency radar system, with a low-frequency of 60 MHz and a high-frequency of 179 MHz, a peak power of 1 kW, and optional pulse widths of 250, 500, and 1000 ns (Cui et al., 2010). The data primarily consist of two radar lines, the locations of which are shown in Fig. 1. The projection coordinate code used is EPSG 3031. The longer line (light blue) extends outward from the highest point of Dome A to form four triangles, with a total length of approximately 160 km, covering most of the 30 × 30 km area around Kunlun. The shorter line (dark blue), starting near Dome A and passing through Kunlun Station, is approximately 50 km long. We refer to these two lines as C21DA and C21KL, respectively.

### 2.2 Age uncertainty of IRHs

In 2007/2008, the Alfred Wegener Institute (AWI) made radar observations that connect the Vostok and Dome A ice cores (Winter et al., 2019). Luo et al. (2022) traced six continuous IRHs from the Vostok ice core to an intersection point in the C21 radar survey along this line, with a maximum age of 160.4 ka. We first leveraged the work of Luo et al. (2023) by directly



applying their tracked depths and established isochrone ages to transfer the Vostok age scale to the AWI radar lines and C21 IRHs, allowing for uncertainty quantification. The two way travel times $t_{IRH}$ converted to depth of IRHs $d_{IRH}$ using

$$d_{IRH} = \frac{c_{ice}t_{IRH}}{2} + z_f \, ,$$  (1)

where the propagation velocity of electromagnetic waves in ice is $c_{ice}$ ( 0.1687 m/ns) and a 15 m firn correction $z_f$ is used (Luo et al., 2022).

The IRH depth uncertainty is influenced by four independent factors, which are combined using a root-mean-square error (RMSE) approach. These factors are: (1) firn correction error, taken as 1.5 m following Luo et al. (2022) for the Dome A region; (2) electromagnetic wave velocity variation, considering a velocity range of 0.168–0.1695 m/ns in ice sheet (Winter et al., 2017), for which the uncertainty is conservatively set to 1% of the IRH depth (Wang et al., 2023b), increasing with depth and corresponding to 8–20 m; (3) radar system range accuracy, with an upper limit of 5 m (Luo et al., 2022); and (4) isochrone crossover point error, reflecting error at the intersection of the C21 and AWI surveys, ranging from 9 to 47 m. Synthesizing these four factors yields a total IRH depth uncertainty along the C21 radar lines of 14–50 m. The total age uncertainty is then obtained from the orthogonal combination of this depth uncertainty and the Vostok ice-core chronology uncertainty (Bouchet et al., 2023), resulting in final age uncertainties for the six IRHs ranging from 1.3 ka for the shallowest layer to 5.8 ka for the deepest layer.

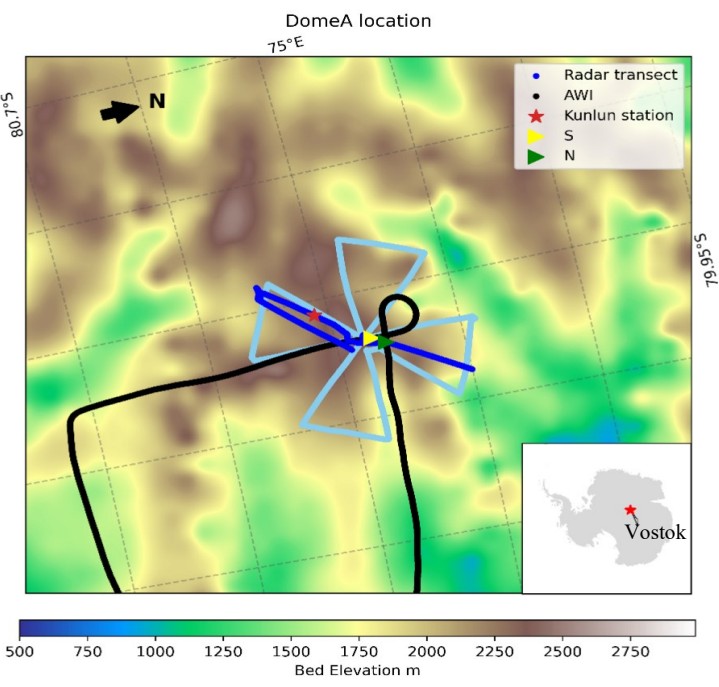



**Figure 1. Location of the Dome A study area. The red star indicates the location of Kunlun Station, and the lines represent radar survey lines. Dark blue is C21DA, and light blue is C21KL. The background color represents bedrock elevation, and the black lines represent surface elevation contours. AWI radar line is shown in black, and its intersections with C21KL are marked by the yellow triangle S (80.3576°S, 77.3796°E) and the green triangle N (80.3420°S, 77.4324°E), respectively.**

## 3 Age Model

To obtain simulated ages for the C21DA and C21KL radar lines of the Dome A region, we used the 1D pseudo-steady-state model modified by Chung et al. (2023) from Parrenin et al. (2006, 2017). The model uses the constraints of the dated IRHs to propagate the age-depth relationships of the nearest ice cores to the study area. This model assumes a 1D pseudo-steady-state, where the geometry and vertical velocity profiles remain constant over time. The modification used is to calculate the basal melt rate using inverted mechanical ice thickness $H_m$ instead of a 1D thermal model The latter can only judge whether the basal ice melts according to the temperature and cannot quantify the melting rate. First, the relationship between the true age $\chi$ and steady state age $\bar{\chi}$ is given by the following equation:

$$d\bar{\chi} = r(t)d\chi \tag{2}$$

$$r(t) = a(x,t)/\bar{a}(x) \tag{3}$$

where $a$ is the accumulation rate, $t$ denotes time, and $\bar{a}$ is the time-averaged accumulation rate. For a given horizontal point $x$, $r(t)$ is derived from accumulation rate and age depth relationships from the Vostok Ice Core (Bouchet et al., 2023). For records older than the Vostok ice core (>408 ka), we assumed $r(t) = 1$, this means that the true age is equal to the steady state age in this case. Second, the steady-state age $\bar{\chi}$ at point $x$ on the radar line can be calculated from the velocity $v(\zeta)$ through the ice layer and the normalized vertical distance differential $d\zeta$, i.e.,

$$\bar{\chi}(\zeta) = \int \frac{1}{v(\zeta)} d\zeta \tag{4}$$

where $v(\zeta)$ is the vertical profile of the vertical velocity, $\zeta = \frac{H_m - d}{H_m}$ is the normalized vertical coordinate (0 at the bottom, 1 at the surface) and $d$ is the ice layer depth. Mechanical ice thickness $H_m$ is the optimized ice thickness parameter. If $H_m$ is less than the actual ice thickness $H$, it indicates stagnant ice; otherwise, it indicates basal melting. We assume no basal melting at the inverted depth,

$$v(\zeta) = \bar{a}\tau(\zeta) = \bar{a}((1-\mu)\omega + \mu) = \bar{a}(1 - \frac{p+2}{p+1}(1-\zeta) + \frac{1}{p+1}(1-\zeta)^{p+2}) \ , \tag{5}$$

where $\tau$ is the thinning function, representing the ratio of the present layer thickness to its original thickness; $\omega$ is the horizontal flux shape function, which controls how the horizontal flux is distributed vertically and approximately equal to the





Lliboutry velocity profile (Lliboutry, 1979); $\mu$ is the sliding factor and $p$ is the shape factor controlling the degree of

nonlinearity of $\omega$ (Lilien et al., 2021). When basal melting m is present, it indicates the value of vertical velocity at bed rock

140

$$m = v\left(\frac{H_m - H}{H_m}\right) \tag{6}$$

To obtain a simulated age-depth relationship closer to the isochronous layer dated at point $x$, we used SciPy least-square

optimization to determine the optimal simulation parameters: $\bar{a}, p' = \ln(p+1), H_m' = \ln(H_m)$. To prevent $p < -1$ and

$H_m < 0$ (Parrenin et al., 2017), we did not directly optimize the parameters $p$ and $H_m$. The cost function $S$ used for

optimization is:

145

$$S = \sum_{1}^{6}\left(\frac{\chi_i^{iso} - \chi^{mod}(d_i^{iso})}{\sigma_i^{iso}}\right)^2 + \left(\frac{p'_{prior} - p'}{\sigma_{p'}}\right)^2 \tag{7}$$

where $\chi_i^{iso}$ is the observed age of the isochronous layer, i is the isochronous layer number, $\chi^{mod}(d_i^{iso})$ is the simulated age at

the corresponding depth, and $\sigma_i^{iso}$ is the age uncertainty of the input dated isochronous layer. Based on previous studies, we

use $p_{prior} = 3, \sigma_{p'} = 1$ (Chung et al., 2023; Wang et al., 2023b). To quantify the reliability of the model, we calculated the

150    standard deviation $\sigma_R$ between the observed and simulated ages:

$$\sigma_R = \sqrt{\frac{R^T R}{n_{iso}}}, \tag{8}$$

$n_{iso}$ is the number of dated isochronous layers, and $R$ is the residual vector at distance point $x$ :

$$R = \frac{\bar{\chi}^{iso} - \bar{\chi}^{mod}}{\bar{\sigma}^{iso}} \tag{9}$$

## 4 Results

155    ### 4.1 Modelled age of C21KL profile

We used six isochronous layers traced along the C21KL survey line as model input. The simulated cross-sectional ages are

shown in Fig. 2. Basal melting is present throughout the entire survey line. Larger simulated ages are primarily found at 4 km,

18 km, and 44 km, corresponding to lower basal melt rates. At Kunlun Station, our simulation results for the dated isochronous



layers are shown in Table 1. The maximum age error is 9.2 ka for IRH6, while the minimum is 0.1 ka for IRH1. The largest

relative age deviation is 9.83% for IRH3, with the remaining percentages being much less than 10%.

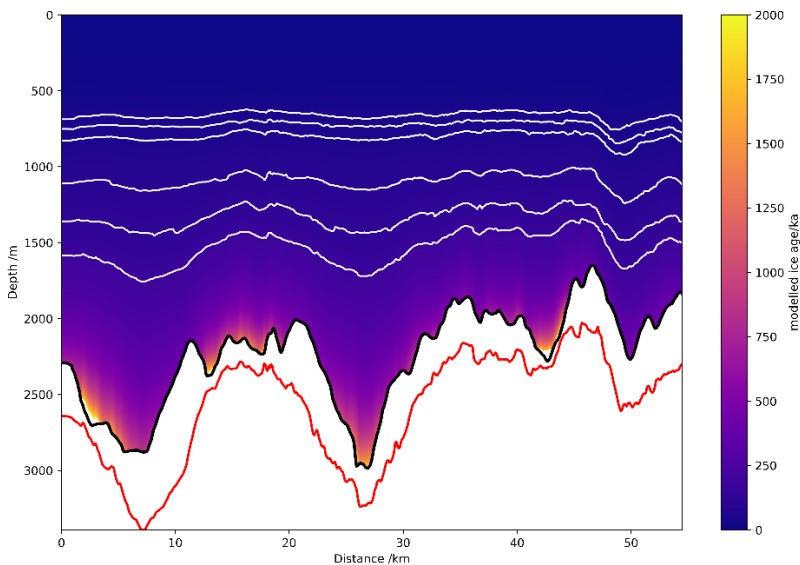

**Figure 2. Cross-section of the C21KL survey line. The black line indicates the observed bedrock position, and the white lines indicate the traced isochronous layers. The red lines indicate the simulated mechanical bedrock depth position. If it is above the bedrock, it indicates that there is stagnant ice at that location; otherwise, there is basal melting.**


**Table 1. Comparison of simulated age and observed age at Kunlun Station**

|       | IRH  | Depth/m  | Modelled age/ka | Observed age/ka | Misfit/% |
|-------|------|----------|-----------------|-----------------|----------|
|       | IRH1 | 601.85   | 35.5            | 35.6            | 0.28     |
|       | IRH2 | 669.52   | 42.1            | 39.5            | 6.58     |
| C21KL | IRH3 | 740.36   | 50.3            | 45.8            | 9.83     |
|       | IRH4 | 1069.83  | 86.2            | 92.9            | 7.21     |
|       | IRH5 | 1296.77  | 123.9           | 121.9           | 1.64     |
|       | IRH6 | 1485.16  | 169.6           | 160.4           | 5.74     |

## 4.2 Basal ice state

Since we use mechanical ice thickness, when the mechanical ice thickness is greater than the actual ice thickness, it indicates

basal melting; otherwise, it indicates the presence of stagnant ice. For IPICS, the Oldest Ice core Project, in order to find older ice, we tend to look for locations with low basal melting or stagnant ice. For paleoclimate reconstruction, parts of the ice core with excessively high age density at the bottom may cause temporal information confusion and render the data uninterpretable



due to disturbances such as shearing and overturning. Here we use a target age density of 20 kyr/m (Fischer et al., 2013; Van

Liefferinge and Pattyn, 2013), consistent with the age density threshold of the Beyond EPICA project at Little Dome C. With

current understanding, our ability to interpret ice cores exceeding this value is limited.

**Figure 3. (A) Maximum simulated ice age, (B) Simulated ice age at 200 m above bedrock, (C) Depth corresponding to the maximum simulated ice age, (D) Age uncertainty of the maximum simulated ice age. The background color represents bedrock elevation, and the black lines represent surface elevation contours.**



In Fig. 3 (A), the maximum ice age ranges from 0.21 to 2.55 Ma, corresponding to a depth range of 1.65 to 2.99 km. Near and north of Kunlun Station, a larger maximum ice age corresponds to greater ice sheet thickness. The simulated maximum ice age at Kunlun Station is $1737 \pm 223$ ka, with an age density of 8.2 kyr/m. In Fig. 3 (D), the maximum age uncertainty is 336.8 ka, with age uncertainties less than 100 ka in most areas. In Fig. 3 (B), the age range at 200 m above the bedrock is 161.8-1176.1 ka, with a spatial distribution similar to that at the ice base (Fig. 3(C)). The age of ice 200 m above the bedrock

age at Kunlun Station is $909 \pm 113$ ka, corresponding to an age density of 2.5 kyr/m.

     Basal melting is a key factor influencing the preservation of old ice at the base of the ice sheet (Fischer et al., 2013). Where basal melting occurs, ice deformation or refreezing can destroy the old ice. The basal melt rate shown in Fig. 4 (B) ranges from 0.01 to 4.2 mm/yr ice equivalent, with the maximum rate exceeding 4 mm/yr, ~28 km northeast of Kunlun. Melt rates are lower in the rest of the survey area, mostly between 0 and 2 mm/yr. The corresponding basal melt rate at Kunlun

Station is 0.14 mm/yr.

     In Fig. 4 (A), The age-density map shows that at 1 Ma, the ice has an age-density range of 2.6 to 6.6 kyr/m, both lower than 20 kyr/m. The figure reveals three potential ice core drilling areas. Area A is located approximately 8 km southwest of Kunlun Station. The age density of the maximum ice age at Kunlun Station in area B is 8.2 kyr/m. Area C is located approximately 5 km north of Kunlun Station. All three areas have low basal melt rates, less than 1 mm/yr. The age density

corresponding to 1 Ma ice is generally higher in Area A and Area C than in Area B.

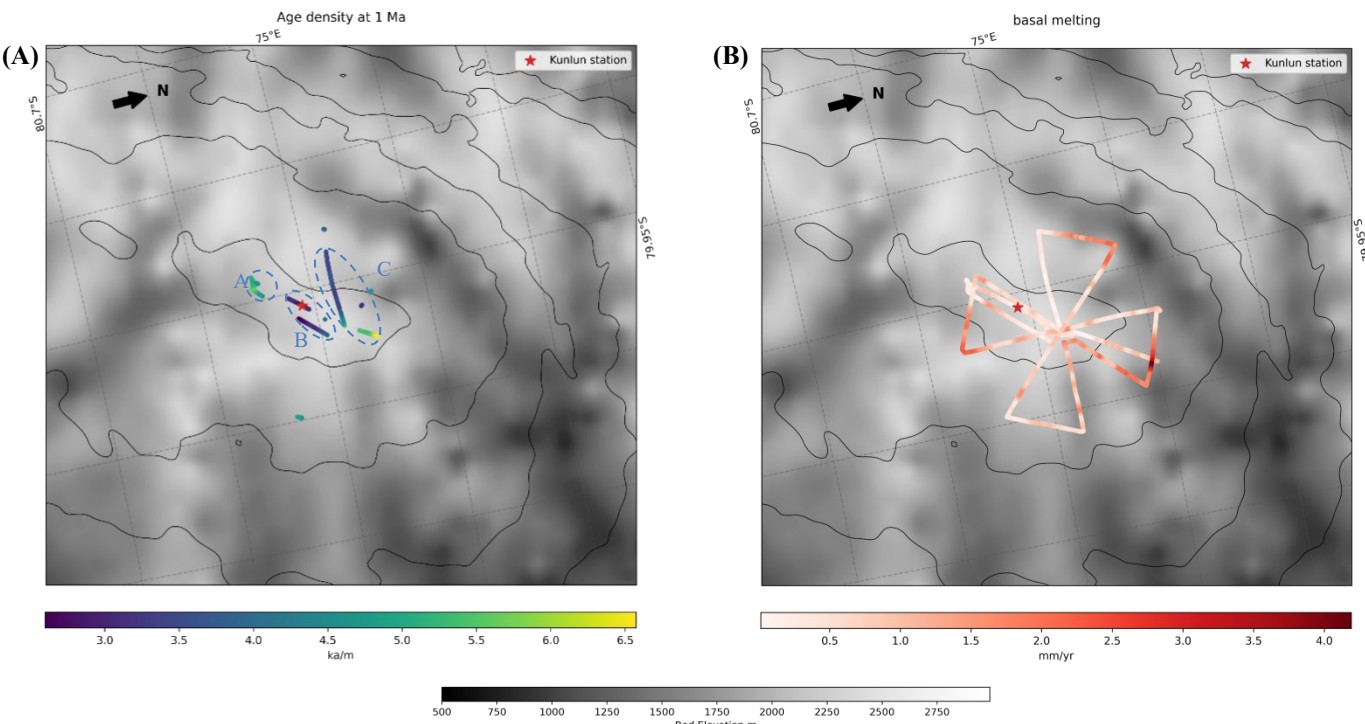



**Figure 4. (A) Age density map at 1 Ma ice age. Dashed lines indicate the optional areas for old ice drilling. A is the 8 km area southwest of Kunlun Station, B is the Kunlun Station area, and C is the 5 km area north of Kunlun Station. (B) Basal melt rate distribution map**

Fig. 5 shows that the standard deviation $\sigma_R$ of the model ranges from 1.5 to 4.4, among which most $\sigma_R$ indicators are greater than 2, which may be related to the horizontal advection in the Dome A region.

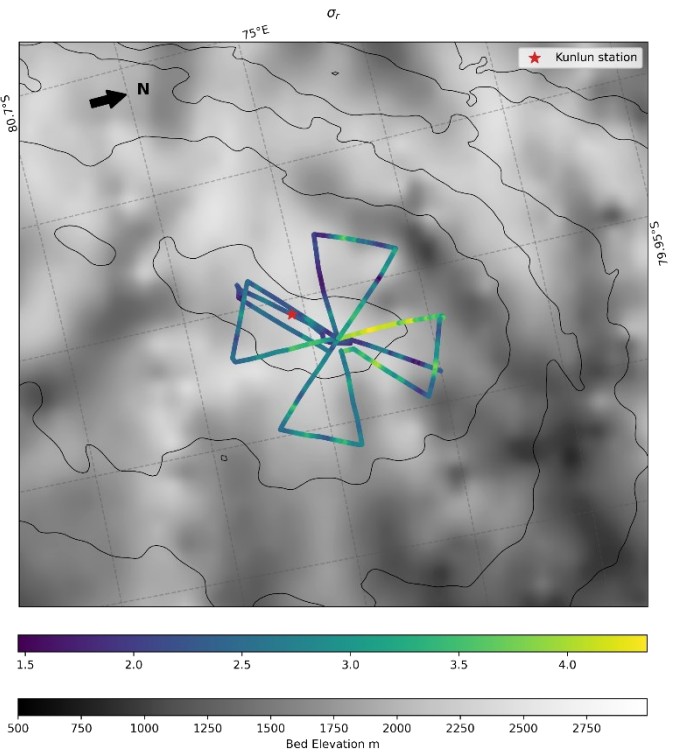

**Figure 5.** $\sigma_R$

### 4.3 Parameters $p$ and accumulation rate $a$

$p$ values and average accumulation rates are important factors influencing the simulated age distribution. In Fig. 6 (A), the average accumulation rate ranges from 24 to 36 mm/yr ice equivalent, with most areas experiencing average accumulation rates less than 32 mm/yr. The average accumulation rate at Kunlun Station is 27 mm/yr, which is close to the maximum of 17-25 mm/yr accumulation rate measured in the Dome A region by Xiao et al. (2008). Higher accumulation rates occur in areas away from the center of the survey line; the $p$ value ranges from -0.74 to -0.14, with a $p$ value of -0.56 at Kunlun Station

(Fig. 6 (B)).



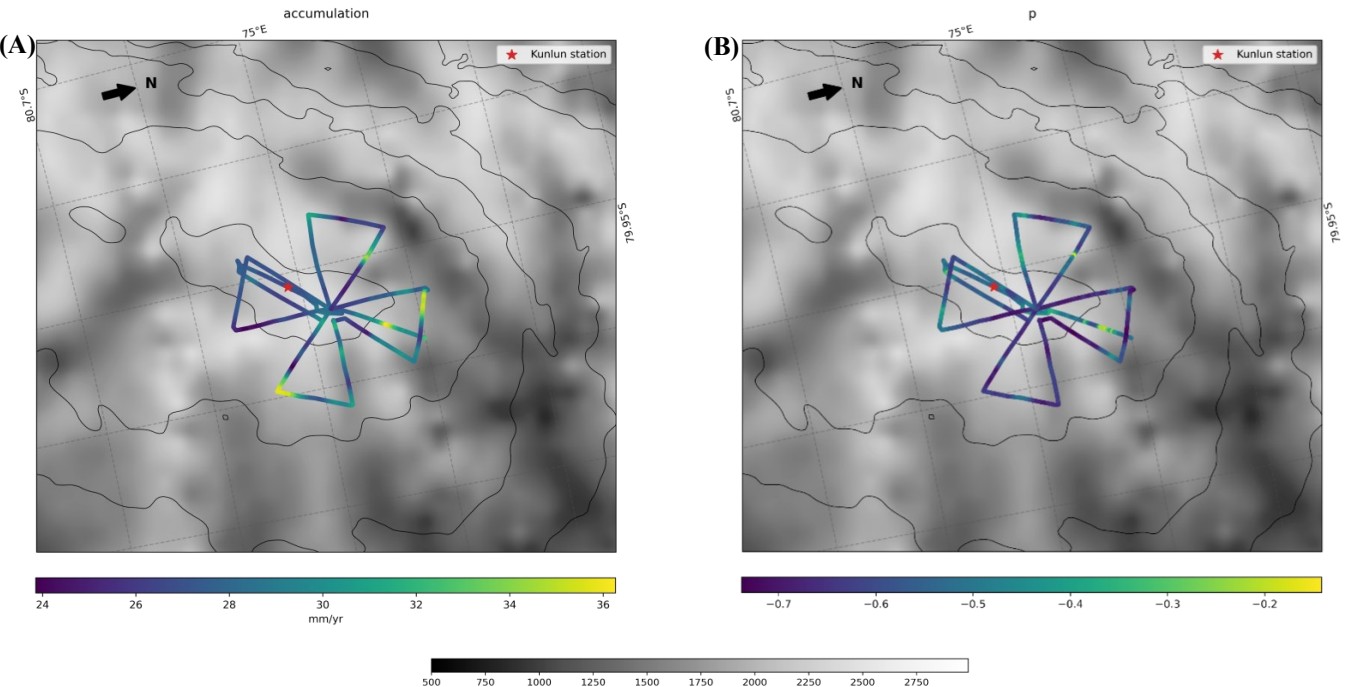

**Figure 6. (A) Simulated steady average accumulation rate, (B) Simulated p-index**

# 5 Discussions

## 5.1 Modelled ice age at Kunlun Station

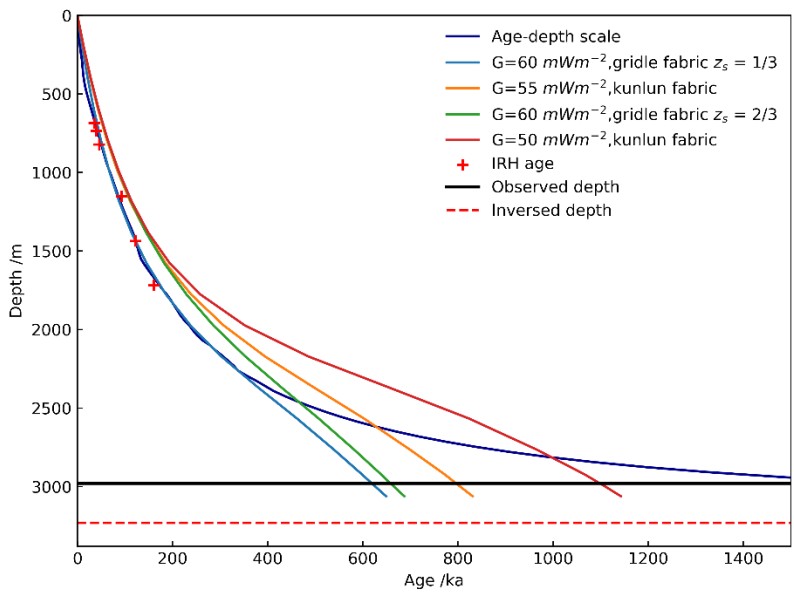



**Figure 7. Comparison of age-depth relationships at Kunlun Station. The red cross marks the observed age, the dark blue line represents the age-depth relationship we simulated, and the other color lines are the curves obtained by Zhao et al. (2018) under simulations of different geothermal flux and fabric combinations. The black curve represents the bedrock, and the red dotted line represents the mechanical ice depth.**

Among existing studies of ice ages in the Dome A region, Sun et al. (2014) used a three-dimensional full-Stokes model to demonstrate that Kunlun Station provides a high-resolution ice core record of 600-700 ka. However, approximately 5 km north of Kunlun Station, an older but lower-resolution ice core record exists. Our simulations also reflect this result. Zhao et al. (2018) used the same methodology to estimate the age of the Dome A region, using data from CHINARE 21 ground-based radar observations within a 30 × 30 km area at Dome A. By varying ice fabric and geothermal flux, their simulated basal

maximum ice age ranges from 650 to 830 ka, corresponding to melt rates of 2-3 mm/yr.

      Compared with the age-depth curve simulated by Zhao et al. (2018) at Kunlun Station, the age ranges of the deep blue curve are similar to Zhao et al. Our simulated curve agrees well with the light blue line at depths above 2500 m (Fig. 7). Below 2500 m depth, our simulated ages increase dramatically with depth, significantly exceeding the age of the curve in light blue. At the bottom of Kunlun, our simulations yield an age of 1737 ± 223 ka, corresponding to a relatively low melt rate of 0.14

mm/yr. This differs significantly from the simulations of Zhao et al. (2018), likely due to the maximum age of our IRHs being only 161 ka, the maximum depth of IRH being less than 60% depth of ice sheet thickness at Kunlun Station, and the lack of IRH constraints at the base. Due to basal melting, it is generally difficult to preserve old ice at the bedrock. In ice cores drilled from Dome C, the oldest ice is found 60 m above the bedrock, while in ice cores from central Greenland, this figure is 200 m. Although our simulated age for the oldest ice at Kunlun Station differs significantly from that of Zhao et al. (2018), our

simulated age of 909±113 ka at a depth of 200 m above the bedrock at Kunlun is considered a conservative estimate of the maximum ice age and is relatively consistent with the maximum ice age simulated by others (Sun et al., 2014; Zhao et al., 2018).

**5.2 Melting rate: comparison with previous studies**

It is generally believed that thicker ice sheets may be more prone to basal melting. With increasing ice thickness, the basal

pressure rises, leading to a reduction in the melting point of ice due to pressure melting. In addition, the thicker ice has a certain insulation effect (Wiskandt and Jourdain, 2025), which may reduce the heat conducted from the bottom of the ice sheet to the atmosphere, resulting in heat accumulation at the bottom of the ice sheet. The cross-section simulations by Chung et al. (2023) at Dome C and Wang et al. (2023b) at Dome Fuji, respectively, also demonstrated this feature.

      However, greater ice thickness doesn't necessarily mean greater melting at the base. When geothermal heat flux is weak

at the base of the ice sheet and the surface temperature is extremely low (for example, in the interior plateau of East Antarctica, where the surface mean temperatures can reach -50°C or lower, and geothermal heat flux is relatively low (Frank, 2010; Turner et al., 2009)), the downward transfer of surface cold can counteract or even exceed the effects of geothermal heating and pressure melting. This may explain why our simulation along the C21KL transect (Fig. 2) did not exhibit the expected trend of higher basal melt rates with increasing ice thickness.




## 5.3 Radar data limitations

In Section 4.1, the simulated IRHs ages at Kunlun Station, when compared to the observed ages, generally show a relatively small bias in the upper layers and a relatively large bias in the lower layers. Wang et al. (2023b) have shown that the number of input IRHs significantly influences model simulation results. Our six input IRHs are fewer than the ~20 used by Chung et al. (2023) in the Dome C region. Depths of IRHs we used here are shallower than those used by Chung et al. (2023) at Dome C and Wang et al. (2023b) at Dome Fuji. The AWI radar system used in the radar line connecting to Vostok has a relatively low resolution, limiting the number of IRHs. This results in a small number of input IRHs connected to C21 radar lines, which in turn affects the simulated ages and uncertainties in the model. Uncertainty in IRHs ages also affects simulated age uncertainties. Our IRHs' depth uncertainty accounts for differences in intersection points' depth, in addition to electromagnetic wave velocity, firn correction, and radar accuracy. Compared to the C21 radar lines' age uncertainty calculated by Wang et al. (2016), our age uncertainty is larger, resulting in an age uncertainty exceeding 200 ka at the bedrock of Kunlun Station.

## 5.4 Model limitations

The modified one-dimensional pseudo-steady-state model proposed by Chung et al. (2023) was used to simulate ice sheet ages in the Dome C and Dome Fuji regions, demonstrating some generalization capabilities. However, the success of the first two models primarily relied on the existence of ice core age-depth relationships in the study area, constrained by isochronous layers traced by radar transects and propagated along radar lines. Furthermore, the model requires that the input radar isochronous layer age be less than the maximum age of the input ice cores. However, the Dome A region is populated by shallow ice cores, whose ages are significantly less than the input isochronous layer age. Therefore, we used the age-depth relationship of the Vostok ice core to interpolate the true age of the Dome A line, which is approximately 700 km away. In previous studies, Chung et al. (2023) used lines up to 200 km from Dome C, and Wang et al. (2023b) lines up to 400 km from Dome Fuji. There is currently no clear limit on the range of radar lines that the model can simulate for input ice cores. Although Vostok and Dome A differ in accumulation rates and ice structure, by adjusting appropriate simulation parameters, our simulated age-depth curves at Kunlun Station generally agree well with the dated isochronous layers.

The deepest isochron depth is a crucial factor influencing model simulation results. The Lliboutry velocity profile is relatively linear in the upper ice sheet, requiring fewer isochron constraints. In the lower part of ice sheet, it's more nonlinear, and deeper IRH constraints are needed to constrain the p-value and thus obtain a reasonable Lliboutry velocity profile. Our existing C21 radar survey isochron depths are primarily shallow, lacking deeper constraints. This is one reason why we will drill ice cores at Dome A.

Fujita et al. (1999) observed that highly anisotropic structures are common in Antarctic ice. Sun et al. (2014) found that variations in ice anisotropy at depth determine the vertical velocity distribution, which in turn affects the age-depth relationship, particularly at deeper layers. Building on this, Zhao et al. (2018) coupled a combination of four unstructured structures and geothermal flux to a three-dimensional full-Stokes model to simulate the age-depth relationship at Kunlun Station. Their



simulated age of the deep ice at Kunlun Station was limited to between 630 and 800 ka. Our model does not directly account for geothermal flux and ice anisotropy, but rather indirectly through an optimized p-value. This may result in a velocity profile shape that is inappropriate for Dome A. Furthermore, the basal melt rate derived from mechanical ice thickness may not be
entirely accurate, as we truncate the velocity profile, meaning some of the value in optimizing p is lost. This ultimately leads to modelled deviations from the actual deep ice age.

Currently, there are still some difficulties in combining the anisotropic 3D Stokes model with observed radar isochronal layer ages. Furthermore, 3D models have some drawbacks, such as the large computational time and observational data requirements, as well as complex boundary conditions. A 2.5D model (Passalacqua et al., 2016) may be a better alternative.
Chung's 2.5D inverse model (Chung et al., 2024), which takes into account the width of the horizontal flow band width, is currently a relatively successful option. However, compared to 1D models, it still requires significantly more observational data.

## 6 Conclusions

We used the 1D pseudo-steady-state model of Chung et al. (2023) to simulate the age of the C21 radar lines. We used six
IRHs with ages derived from the Vostok ice core as constraints. Based on these data, we propagated the age-depth relationship of the Vostok ice core to the C21 radar survey in the Dome A region. This method obtained parameters such as the inferred ice age and accumulation rate along the radar lines.

Based on the simulation results, we have identified three regions of ice over 1Ma: Kunlun Station, ~8 km south of Kunlun Station, and ~5 km north of Kunlun Station. Although these regions exhibit higher age uncertainty than others, they are
characterized by greater ice thickness and lower basal melt rates (<1 mm/yr), while the age density remains below 20 kyr/m. The maximum ice-age density in both the southern and northern regions adjacent to Kunlun Station exceeds that simulated at the station itself, with particularly elevated values in the northern sector. Notably, this northern sector coincides with the promising old-ice area previously identified by Zhao et al. (2018). While our simulated ages at Kunlun Station partially overlap the 650–830 ka range reported by Zhao et al. (2018), our conservative estimate approaches 1 Ma and is accompanied by
reduced uncertainty. These differences are likely due to the simplified yet effective modelling framework adopted here, which allows the integration of more spatially explicit radar-derived age constraints.

In summary, our observation-constrained estimates indicate that both the Kunlun Station region and the adjacent northern region have strong potential to preserve ice older than 1 Ma, with the northern region showing older ice and substantially higher age density. These results provide practical guidance for prioritizing promising old ice location near Dome A and
contribute to ongoing international efforts to recover ice over 1 Ma.



*Code availability.* The model code is available on Github https: //github.com/ailsachung/IsoInv1D  (last access: 9 August
2025; DOI: https://doi.org/10.5281/zenodo.8189792, Chung and Parrenin, 2023).

*Data availability.* We used the surface elevation data from REMA surface data (https://fridge.pgc.umn.edu/ (Howat et al.,
2019)) and bed elevation from  BedMachine v3 (https://nsidc.org/data/nsidc-0756/versions/3 (Morlighem et al., 2020)). The
CHINARE 21 radar data used in this study is available at https://www.mdpi.com/2072-4292/15/7/1726 (Wang et al., 2023a).
The AWI radar data is available at https://ieeexplore.ieee.org/document/9904613 (Luo et al., 2022).

*Author contributions.* TX designed the experiments. SB collected the radar data. LD performed the simulations and wrote the
drafts. TX, AC, FP provided revision suggestions. All authors commented on and edited drafts of this paper.

*Competing interests.* The authors declare that they have no conflict of interest.

*Acknowledgments.* We acknowledge the support from the National Natural Science Foundation of China under Grant
(42276257), and the Top-notch Project under the 2024 Shanghai Oriental Talents Program (BJKJ2024035). The authors thank
the Chinese National Antarctic Research Expedition for their help in the field data collection.

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
