# Peer review of "Simulated evidence for ice over 1Ma in the Dome A region, East Antarctica"

_EGUsphere, 2025_

## Referee Comment (RC1)

**Review of Li et al. (2026) 'Simulated evidence for ice over 1Ma in the Dome A region, East Antarctica'**

**Summary**

In this paper, the authors applied a 1D pseudo-steady-state model of Chung et al. (2023) to Dome A region, and found older ice age at the deep depth than an earlier study Zhao et al. (2018), which used a 3D full-Stokes model. They also identified three potential locations where ice older than 1 Ma may be reserved. The result may be exciting for ice core drilling program. But it need to be emphasized that the depth of the isochronous layers are shallow, and the key accumulation rate and age depth relationship in the model is from Vostok ice core, which may bring large uncertainty to this result.

The method has been used in Dome C and Dome Fuji. But we must be careful that optimization parameters obtained under limited data constraints may be far from the true optimal values. I am wondering how the modelled age at Kunlun station would change if another accumulation rate and age depth relationship is taken? For instance, use the accumulation rate and age depth relationship at the Dome Fuji or Dome C.

Beside, the wording requires refinement, the figures should be improved. I have lots of minor comments for the figures.

**Major Comments**

As the authors said, they took the accumulation rate and age depth relationship from Vostok ice core which is 700 km away from Kunlun station. It may have large difference from the real relationship at Kunlun station. Hence it may cause large uncertainty in the optimal value of parameter $p$ and the modelled age, as it is a key function in this 1D pseudo-steady-state age model. So I am wondering how the modelled age at Kunlun station would change if another accumulation rate and age depth relationship is taken? For instance, use the accumulation rate and age depth relationship at the Dome Fuji or Dome C. I also would like to see the plot of normalized vertical velocity. Please add this plot at Kunlun station. Also, it would be interesting to compare the normalized vertical velocity with Zhao et al. (2018).

**Minor Comments**

Line 1, what do you mean by 'high precision'? Why do you think your model is a high precision model?
Line 19, there are two commas.
Line 21, add 'of' after the last 'age'.
Line 24, It need to point out the model limitation, which may bring large uncertainty to this result.
Line 25, 'Isoinv1D model' is only used here. But you use '1D pseudo-steady-state model' in section 3. Use the same name.

Line 37, change 'simulations' to 'projections'

Line 52, what is 'NP'? Provide its full name when it first appears.

Line 56, it is hard to understand this sentence '... the maximum ice age is defined as ice characterized by an age density of 20 kyr/m.'

Line 65, 'a full Stokes coupled heat equation model' is wrong description. Full Stokes model includes the heat equation.

Line 70, it is not clear what do you mean by 'these factors'. The main uncertainty comes from GHF. 'greater uncertainty', why do you use 'greater', than what?

Line 71, Provide the full name of 'IRH' when it first appears.

Line 73, 'supportf' is a typo

Line 85-87, avoid to use 'light blue' and 'dark blue' in the main text. They should only appear in figure captions. Remove them. Use C21DA and C21KL instead.

Line 92, is it Luo et al. (2023) or Luo et al. (2022)?

Line 94-96, for some reason, the font size of the subscripts for the variables in the text and in the equations is too large.

Line 120. It is not correct that '... cannot quantify the melting rate.'

Line 126, why do you make this assumption?

Eqn (4), check if you missed $H_m$.

Eqn (5), use parentheses that match the formula height.

For all the equations, double check the variables if they are scalar or vector. Vectors should be in bold.

Page 6, the subscript of $H_m$ in this page looks odd.

Line 157-158, what do they mean by '4 km, 18 km, and 44 km'? Are they distance from somewhere? Please describe them accurately.

Line 181. I do not understand this sentence 'Near and north of Kunlun Station, a larger maximum ice age corresponds to greater ice sheet thickness. '.

Line 184, I assume you mean Fig. 3 A?

Line 223, add 'as Sun et al., (2014)' after 'the same methodology'.

Line 226-228, avoid to use words like 'blue curve' in the main text. Replace them with scientific description.

Line 230-232. Rephrase this long sentence.

Line 233. There is something wrong in this sentence. 'this figure is 200 m'?

Line 248-249. I do not agree with this sentence. Do you have similar cold surface temperature and low geothermal heat flux in Dome A?

Line 291, do not use subjective words such as 'successful'.

Line 294-310. Please rephrase the conclusion paragraph. The wording needs to be improved. For instance, you cannot say "The method obtained parameters". Is this '8 km south' correct? It is said '8 km southwest' in section 4.2.

Line 305. Rephrase this sentence 'These differences are likely due to the simplified yet effective modelling framework adopted here, which allows the integration of more spatially explicit radar-derived age constraints'. Both this study and Zhao et al. (2018) used the same radar-derived age constraints. Maybe you write 'The variation in results is mainly due to different methods.'

**For the figures:** There are only one or two numerical values on the longitude and latitude lines in the maps. It is not easy to read the map. For all the maps, add numerical values to all the longitude and latitude lines. Besides, for all the maps, add the scale bar. The text and numbers on the figures need to be enlarged.

**Figure 1.** I would label C21DA for the dark blue line and C21KL for the dark blue line, to be consistent with what you said in the caption. It is hard to see the two intersections of AWI with C21KL. Are the two points 'S' and 'N' necessary to show? Give the reason why you need to label them. If you need to, I suggest to use other letter to represent them, because S and N can easily be confused with south and north. I do not think it is correct '...and the black lines represent surface elevation contours.' There are no surface elevation contours. You did not say anything for the inset figure. The label of 'Vostok' could be smaller. There are black lines in the inset box, what are they? Describe it in the figure caption. The sub-caption of the figure is not need, remove it. Put the unit 'm' into parentheses.

**Figure 2.** It is hard to match this figure with figure 1. You have stretched the four triangles into a straight line. Where is your starting point? And the direction? Where is Kunlun station? Mark it.

**Figure 3-6.** Much of the figure domain is useless and not interesting; the spatial extent of the plot can be reduced. This legend for 'Kunlun station' is not needed. Say it in the caption and remove the red star legend. Remove the sub-captions of the figure. Please them at the colorbar. Put the units into parentheses. Chang the units like 'mm/yr' to 'mm yr$^{-1}$'. The text and numbers on the figures need to be enlarged. Circle the same optional area in Figure 4b as in Figure 4a.

The figure 4 caption is unclear, such as 'A is the 8 km area southwest of Kunlun Station, B is the Kunlun Station area, and C is the 5 km area north of Kunlun Station.'.

Please improve it, to be consistent with the main text. You may say 'Area A is 8 km southwest of Kunlun Station, ...'. You can combine figure 4 and 5 into one figure.

**Figure 7.** Replace the legend 'Age-depth scale' with 'this study'. Add a subplot to show the vertical velocity.